# A Population Census of Large Herbivores Based on UAV and Its Effects on Grazing Pressure in the Yellow-River-Source National Park, China

**DOI:** 10.3390/ijerph16224402

**Published:** 2019-11-11

**Authors:** Fan Yang, Quanqin Shao, Zhigang Jiang

**Affiliations:** 1School of Economics and Management, Zhejiang Ocean University, Zhoushan, 316022, China; yangf.16b@igsnrr.ac.cn; 2Key Laboratory of Land Surface Pattern and Simulation, Institute of Geographic Sciences and Natural Resources Research, Chinese Academy of Sciences, Beijing, 100101, China; 3Institute of Zoology, Chinese Academy of Sciences, Beijing, 100101, China; jiangzg@ioz.ac.cn

**Keywords:** UAV remote sensing, large wild herbivore populations, forage yield, functional zones, forage-livestock balance

## Abstract

Using the Yellow-River-Source National Park (YRSNP) as a study site, an unmanned aerial vehicle (UAV) remote sensing and line transect method was used to investigate the number of wild herbivorous animals and livestock, including the kiang (*Equus kiang*) and Tibetan gazelle (*Procapra picticaudata*). A downscaling algorithm was used to generate the forage yield data in YRSNP based on a 30-m spatial resolution. On this basis, we estimated the forage–livestock balance, which included both wild animals and livestock, and analyzed the effects of functional zone planning in national parks on the forage–livestock balance in YRSNP. The results showed that the estimates of large herbivore population numbers in YRSNP based on population density in the aerial sample strips, which were compared and validated with official statistics and warm season survey results, indicated that the numbers of kiangs and Tibetan gazelles in the 2017 cold season were 12,900 and 12,100, respectively. The numbers of domestic yaks, Tibetan sheep, and horses were 53,400, 76,800, and 800, respectively, and the total number of sheep units was 353,200. The ratio of large wild herbivores and livestock sheep units was 1:5. Large wild herbivores have different preferences for functional zones, preferring ecological restoration areas consisting mainly of sparse grassland. The grazing pressure indices of the core reserve areas and ecological restoration areas were 0.168 and 0.276, respectively, indicating that these two regions still have high grazing potential. However, the grazing pressure index of the traditional utilization areas was 1.754, indicating that these grasslands are severely overloaded. After the planning and implementation of functional zones, the grazing pressure index of YRSNP was 1.967. Under this measure, the number of livestock was not reduced and the grazing pressure nearly doubled, indicating that forage–livestock conflict has become more severe in YRSNP.

## 1. Introduction

The Yellow-River-Source National Park (YRSNP) is located in the hinterland of the Tibetan Plateau at the source of the Yellow River, and is an important supply area for freshwater resources in China [1]. This region is one of the richest biodiversity areas and constitutes a sensitive zone for climate change in the Northern Hemisphere as well as globally [2,3]. Its unique geographical location, rich natural resources, and important ecological functions have enabled it to become an important ecological security barrier in China [4,5]. However, overgrazing, grassland degradation, seasonal imbalances, and other developmental bottlenecks in the traditional animal husbandry industry in this region have severely threatened the maintenance of grassland ecosystem services and functions [6,7,8]. This location is an ideal example of this conflict. The establishment of the national park system has enabled an effective conservation of wild animals in the park and facilitated the rapid recovery of population numbers, which has resulted in increased competition with livestock for forage grass, and has even had some impact on the local grassland ecosystem and animal husbandry industry [9]. However, the number of large wild herbivore populations, their distribution patterns in the region, and their effects on the animal husbandry industry are still unknown. The effectivity of wild animal conservation in the region is thus unclear.

Survey methods for large herbivore populations include ground surveys and remote sensing surveys. Ground surveys are simple and easy to carry out but have high costs, are time-consuming, and cannot be frequently conducted in a large area. In addition, the habitats of herbivorous animals are usually in rugged terrain that is inaccessible. Sampling survey results cannot accurately reflect the number and variation rate of animal habitats and often lacks precise spatial positional information [10,11]. Remote sensing surveys mainly employ manned planes and unmanned aerial vehicles (UAVs) for aerial photography to record individuals. Of these two methods, the UAV platform has a lower cost, more flexible operation, and provides a high resolution, which has rendered it a new and effective method for studying animal diversity [12]. In terms of herbivore population size estimation, the time cost of UAV surveying is far lower than that of ground surveying. Currently, UAV-based animal surveys mainly focus on examining the possibility and detection rate of species recognition as well as the factors affecting species recognition. Animals that have been monitored using UAVs include caribou [13], elephants in Western Africa [14], rhinoceroses [15], and black bears [16], but the survey areas in these studies were all smaller than 30 km^2^. In addition to the direct monitoring of animal data and distribution, UAVs can enable detailed access to habitat information for habitat studies of different animals, such as koalas and chimpanzees [17,18]. Some studies have also evaluated the disturbances caused by UAV monitoring on wild animals [19,20].

Research on the forage–livestock balance has mainly focused on pasture yield, livestock feed intake, pasture utilization rate, and livestock carrying capacity calculations [21,22,23,24,25,26]. Thapa et al. [27] used livestock numbers to calculate the livestock carrying capacity of land resources in the southern mountains of Nepal. Silori et al. [28] noted that the increasing number of livestock in southern India has disturbed the habitat of large mammals in the Mudumalai Wildlife Sanctuary. Zhang et al. [29] used remote sensing to estimate the forage yields and livestock carrying capacities of different types of grasslands in the Three-River Source region and found that there were 6.52 million overgrazing sheep units in that region in 2010. Fan et al. [30] used the Global Production Efficiency Model (GLO-PEM) model to calculate forage yield and grazing pressure, and results indicated that climate change is the main factor affecting forage yield in the Three-River Source region, with different grazing management models similarly affecting forage yield. Cai et al. [31] showed that ecological livestock reduction projects in Madoi County have reversed grassland degradation in some areas. There are few reports on the effects of ecological protection and management policies on the forage–livestock balance in grasslands with high numbers of wild herbivores in China.

Within the YRSNP, pasture is at the supply end, while large wild herbivorous animals and human grazing activity are at the consumption end of the spectrum. These three factors reflect the coupled human–land relationship of the supply and consumption of ecosystem services. The quantitative characterization and elucidation of the coupled relationship between grasslands, wild animals, and livestock populations are required for the long-term sustainability of Tibetan gazelles for both the national park and local residents. In this study, UAV aerial photography, ground surveying, and other techniques were used to estimate the number of large herbivore populations inside the park and to analyze their distribution patterns. This was done to facilitate the scientific and rational conservation of wild animals, obtain accurate estimates of the forage–livestock balance (which includes wild animals and livestock), and analyze the effects of relevant policies and plans on the forage–livestock balance. This will allow for the effective adjustment of grassland resource utilization plans and will provide a scientific basis for the maintenance of grassland ecosystem cycles, thus promoting the construction of nature reserves and national parks in the Three-River Source region.

## 2. Materials and Methods

### 2.1. Study Area

The YRSNP is located in Madoi county, Golog Tibetan Autonomous Prefecture, Qinghai province. The longitude is 97°1'20"–99°14'57" E and the latitude is 33°55'5"–35°28'15". The study area includes the two protection zones of Gyaring Lake-Ngoring Lake and Xingxing Sea in the Three-River Source National Nature Reserve and encompasses an area of 19,100 km^2^, accounting for 78.01% of the total area of Madoi county. The terrain slopes from the northwest to southeast with a mean altitude of around 4400 m, an annual mean temperature of −4 °C, an annual mean precipitation of 303.9 mm, an annual evaporation of 1374.6 mm, and an annual relative humidity of 38%. The study area has a classical alpine steppe climate. The park is dominated by rivers, and the course of the Yellow River is approximately 408 km. There are numerous lakes in the park, of which the Gyaring Lake and Ngoring Lake are the two largest natural lakes at the upper reaches of the Yellow River, forming a “thousand lakes” landscape with different lakes in the region. The park consists mainly of alpine meadows. *Kobresia pygmaea*, *Kobresia graminifolia*, and *Kobresia humilis* are the dominant plant species. There are 69 key conserved animal species in China, of which nine (such as the snow leopard, Thorold's deer, kiang (*Equus kiang*), and wild yak (*Bos grunniens mutus*)) are Class I protected animals [32].

The YRSNP is divided into core reserve areas (8600 km^2^), ecological restoration areas (2400 km^2^), and traditional utilization areas (8100 km^2^). The level of conservation decreases in this order, while the degree of utilization and public access increases in this order (Figure 1).

### 2.2. Estimation of Large Herbivore Populations

#### 2.2.1. UAV Survey

The design of the UAV aerial photography sample strips was based on the *Code of Practice for Terrestrial Wildlife and Its Habitat Survey* issued by the State Forestry and Grassland Administration and comprehensively considered terrain, land use/cover, and vegetation types [33]. Systematic sampling was used to uniformly distribute the survey sample strips within YRSNP (Figure 2). Surveys were conducted in the 2016 warm season and 2017 cold season. The sampling intensity of the two UAV surveys conformed to the specifications of various departments.

Two self-developed UAVs were used for UAV aerial photography in the 2016 warm season, of which the diesel-powered UAV was used seven times and the electric-powered UAV was used twice. A Sony ILCE Lens-Style camera was mounted on different UAVs. The speed of these UAVs was 108 and 72 km/h, respectively. The relative height of flight was 700 m, and the resolution was 15–18.5 cm. A total of 23,810 photographs were taken and the area covered was 2728 km^2^. In the 2017 cold season, the self-developed electric-powered UAV and the Feima Robotics F1000 electric-powered UAV was used for UAV aerial photography. The height used was 200–350 m, and the resolution was 4–7 cm. A total of 23,784 photographs were taken and the area covered was 350 km^2^. The fore-and-aft overlap and side overlap of the two aerial photography sessions were 80% and 60%, respectively.

LiMapper, Pix4Dmapper, and Feima Image Stitching software were used for the stitching of aerial photography images. Further, human–computer interaction (HCI) was used to interpret the stitched images. The HCI method means that the professional interpreters in the aspect of remote sensing and herbivores identify the message of remote sensing using geographic information system software. The seven elements of remote sensing interpretation (shape, size, hues, shadows, images, layout, texture) were combined with animal activity patterns and characteristics that were observed in the field survey for summarizing and constructing an applicable interpretation marker library. The procedure for HCI interpretation is as follows: the interpretation group identifies animals from the stitched images and carries out recognition based on the interpretation marker library, constructs vector points for animal categories, and simultaneously identifies raw images for inspection. The quality inspection team overlaps the identified vector points and stitched images before searching for the raw images for re-verification.

#### 2.2.2. Line Transect Survey

In order to validate the reliability of the UAV survey results, the line transect method was used for synchronous ground surveys on kiangs and Tibetan gazelles (*Procapra picticaudata*) at the two sides of the flight route of the UAVs. The HCIYET HT-1500A rangefinder and compass were used to measure the vertical distance of animals to the survey route, and the following variables were recorded: species of wild animals, group size, geographical coordinates, vertical distance to the survey route, survey time, and route length. In order to reduce the errors caused by distance on the observations, this survey only recorded observations within 500 m of the survey route. Individuals with a distance less than 100 m were considered to be in the same group.

The total length of the ground survey route was 304.07 km, of which 4.52 km was surveyed by foot and 299.55 km was surveyed by vehicle. The area of the sampling survey was 304.07 km^2^, accounting for 1.37% of the area of YRSNP. The sampling survey intensity conformed to the requirements of not less than 1% stated in the *National Terrestrial Wildlife Resource Survey and Monitoring Technical Guidelines*. Figure 2 shows the ground survey route.

Ground survey data were collected and compiled, and the number of each species of wild animal was obtained. The following equation was used to calculate the mean density of each type of wild animal [34]:(1)$$D=\;\frac{n}{2LW}$$ where *n* is the number of animals in the sampling survey, *L* is the length of the survey route, and *W* is the unilateral average observation distance.

The number of wild animal populations in the entire region was calculated using the following equation:(2)N= DS where *S* is the total habitat area of the target animal species (except for lakes and buildings).

#### 2.2.3. A Statistical Method Based on the Sampling Survey

The animal vector points obtained through human–computer interaction can be used to determine the location and quantity of different herbivores within the UAV sample strips. In this study, we designed a statistical method for calculating the number of animals in the region based on the sample strips that was based on the assumption that wild animals seek cooler, high-altitude (usually cold season pastures) regions with abundant forage and less human interference during the warm season as habitats for feeding, and low-altitude, low-lying warm terrain (usually warm season pastures) during the cold season for shelter and feeding [35]. We used pasture distribution data of cold and warm seasons to estimate the number of large herbivore populations within the YRSNP. The relevant formula is as follows [36]:(3)$$Q=\;\overset{5}{\underset{i=1}{\sum }}\overset{2}{\underset{j=1}{\sum }}K_{i,j}R_{j}$$ where Q represents the total number of large herbivores in YRSNP; *K**_i, j_* represents the population density (individuals/km^2^) of the *i*^th^ herbivore in the *j*^th^ grassland type within the sample strip; and *R**_j_* represents the area of the *j*^th^ grassland utilization type in YRSNP (km^2^).

### 2.3. Estimation of Forage Yield and the Downscaling Algorithm

Remote sensing empirical models were used to estimate the forage yield in YRSNP. Taking into account the large differences between the different types of grasslands, the grasslands in YRSNP were divided into alpine meadows and alpine steppes to construct five empirical models of normalized difference vegetation index (NDVI) and forage yield; namely, linear function, logarithmic function, power function, exponential function, and quadratic polynomial function [37].

In this paper, the 500-m resolution NDVI product in the MOD13A1 dataset was used every eight days for maximum synthesis. The 393 quadrat datasets of field forage yield measurements (2011–2015) from end-July or end-August every year and the NDVI maximum synthesis data (MOD13A1) in the years corresponding to the extracted ground quadrat data were used to construct the models. After an F test was carried out on the regression model, precision comparison was performed (Table 1). The coefficients of correlation of the five models were used to select the best model for the estimation of the annual forage yield in the Three-River Source grassland, and this model was applied to the YRSNP region [38].

Although the 500-m-resolution forage yield data generated from the empirical model can adequately reflect forage yield status on a large scale, the data cannot precisely show the actual status of different types of coverage in small regions. Therefore, the medium-resolution forage yield data and high-resolution 30-m vegetation data were fused to obtain 30-m forage yield data. The specific steps were as follows. First, wave band calculations were used to obtain NDVI data (Landsat8-TM images). Following that, the binary pixel model was used for the estimation of vegetation coverage. Then, high-resolution-based land cover data were used to extract pure pixels (single 500 × 500 m land cover grid) to calculate the forage yield and vegetation cover within the pure pixels, extract 54 pairs of data, and fit the relationship curve of the two. Finally, the 30-m high-resolution forage yield data were calculated pixel by pixel (Figure 3).

### 2.4. Selection Coefficient and Selection Index

In this paper, the selection coefficient W*_i_* and selection index E*_i_* were used to measure the preferences or avoidance of Tibetan gazelles, kiangs, and other large wild herbivores towards different functional zones. The calculation method was as follows [39]:(4)$$W_{i}=\;\frac{\frac{r_{i}}{p_{i}}}{\sum \frac{r_{i}}{p_{i}}}$$
(5)$$E_{i}=\;\frac{W_{i}-\frac{1}{n}}{W_{i}+\frac{1}{n}}$$ where W*_i_* is the selection coefficient, E*_i_* is the selection index, *i* refers to a specific environmental characteristic, *r**_i_* is the number of quadrats containing the *i*^th^ characteristic that is selected by the species, *p**_i_* is the total number of quadrats containing the *i*^th^ characteristic, and *n* is the number of grades for a specific environmental characteristic (*n* = 1, 2,… n). When E*_i_* = 1, the species has a particular preference; E*_i_* = −1, there is no selection; E*_i_* < −0.1, there is negative selection; E*_i_* > 0.1, there is positive selection; and E*_i_* = 0, there is random selection. There is random selection when −0.1 ≤ E*_i_* ≤ 0.1.

### 2.5. Calculation of Grazing Pressure

In order to analyze and evaluate the forage–livestock conflict characteristics in different functional zones in YRSNP, we calculated the grazing pressure for grasslands based on livestock and wild animals (kiangs and Tibetan gazelles). Table 2 shows the specific classification criteria for the grazing pressure index in Qinghai province [40].

The calculation equation for the grassland grazing pressure index proposed by Fan et al. [41] is as follows:(6)$$I_{p}=\;\frac{C_{p-a}}{C_{p-t}}$$
(7)$$C_{p-a}=\;\frac{C_{n}}{A}$$
(8)$$C_{p-t}=\;\frac{Y\times K\times U}{R\times T}$$ where I*_p_* is the grassland grazing pressure index; *C**_p-a_* and *C**_p-t_* are the actual livestock carrying capacity and theoretical livestock carrying capacity of the grassland (sheep units/hm^2^), respectively; C*_n_* is the number of livestock and wild animals (sheep units) that were observed by the UAV; A is the usable area (hm^2^) of the grassland in the study site; *Y* is the forage yield per unit area (kg/hm^2^); K is the proportion of edible forage grass and has a value of 0.6 (based on actual measurement data in the YRSNP); U is the utilization rate of grasslands and has a value of 0.65; and *R* is the daily food intake of a standard sheep unit animal (1.8 kg DM/day). Table 3 shows the standard sheep unit conversion method for various herbivores [42]. T is the number of days of grazing and was calculated based on 365 days of grazing.

## 3. Results

### 3.1. Distribution Characteristics of Large Herbivore Populations

#### 3.1.1. Survey Results of the UAV Sample Strip

In the 2017 cold season, a total of 2248 large herbivorous animals were found within the UAV aerial photography sample strip, which corresponded to 6135.5 sheep units. Among these animals, 1036.5 sheep units belonged to large wild herbivorous animals and 5099 sheep units belonged to livestock. The ratio of these two types of animals was 1:5. There were 252 kiangs, which corresponded to 1008 sheep units at a density of 1.21 individuals/km^2^. There were 57 Tibetan gazelles, which corresponded to 28.5 sheep units at a density of 0.27 individuals/km^2^. There were 1030 domestic yaks, which corresponded to 4120 sheep units at a density of 4.95 individuals/km^2^. There were 895 domesticated Tibetan sheep, which corresponded to 895 sheep units at a density of 4.30 individuals/km^2^. There were 14 horses, which corresponded to 84 sheep units at a density of 0.07 individuals/km^2^. The details are shown in Table 4. Large wild herbivores accounted for 13.75% of all large herbivorous animals. The large wild herbivorous animal density results in the UAV aerial photography sample strips were generally consistent with the first terrestrial wild animal resource survey results obtained by the State Forestry and Grassland Administration in 1995–2003 wherein the densities of kiangs and Tibetan gazelles were 0.87 and 0.56 individuals/km^2^, respectively, in Qinghai province [43].

#### 3.1.2. Line Transect Results

Numbered lists can be added as follows: In the simultaneously performed ground sample strip survey, a total of 368 large wild herbivorous animals were found, of which there were 236 kiangs in 30 populations and 132 Tibetan gazelles in 24 populations. The survey route had a total length of 304.07 km, the survey sample strip width was 1000 m, and the total survey area was 304.07 km^2^. Based on the calculations in Equation (1), the mean densities of kiangs and Tibetan gazelles in YRSNP were 0.78 and 0.43 individuals/km^2^, respectively (Table 5).

The kiang and Tibetan gazelle exhibited different population characteristics: Most kiang populations had 2–9 individuals, accounting for 76.67% of all populations. The populations with the highest frequencies had 2 individuals, of which there were 9 such populations. The mean number of individuals per population was 7.86 individuals. Conversely, most Tibetan gazelle populations contained 2–12 individuals, accounting for 91.67% of all populations. The populations with the highest frequency contained 4 individuals, and the mean number of individuals per population was 5.5 individuals.

The 9, 15, and 16 April 2017 ground survey routes were synchronous with some of the UAV sampling sites. Of these surveys, the ground synchronous survey on 9 April differed from the UAV survey by 1 h. The ground survey found 3 kiang populations, containing 31, 51, and 30 individuals, and the total number of individuals was 112. The UAV survey found 3 kiang populations, containing 28, 48, and 31 individuals, and the total number of individuals was 107. The difference between the two was small, and the errors of the UAV survey results when compared with the ground survey results were 9.68%, 5.88%, and 3.33%. The overall error when the UAV survey results were compared with the ground survey results was 4.46%. The 15 and 16 April ground synchronous surveys also differed from UAV surveys by 1 h, but no animals were found. The UAV survey results were the same as the ground survey results.

#### 3.1.3. Estimation and Verification of Large Herbivore Populations

Numbered lists can be added as follows: The cold and warm season pasture estimation method considers both the activity patterns of livestock during grazing in cold and warm season pastures and the lifestyle habits of wild animals. Relevant studies have shown that kiangs tend to migrate to low-lying areas in winter and higher places in summer, and high elevation (>4200 m) is typically used as a basis for delineating cold and warm season pastures (Figure 4).

Equation (3) was used for the calculation. The results showed that there were 53,400 domestic yaks, 76,800 Tibetan sheep, and 800 horses during the 2017 cold season in the YRSNP. There were 12,900 kiangs and 12100 Tibetan gazelles. From this, it is evident that the number of large wild herbivorous animals was still smaller than livestock, as the former was only 19.09% of the latter. After conversion to sheep units, it was estimated that there were 353,200 sheep units of large herbivorous animals in YRSNP, of which there were 213,500, 76,800, 5300, 51,600, and 6000 sheep units of domestic yaks, Tibetan sheep, horses, kiangs, and Tibetan gazelles, respectively. The ratio of large wild herbivorous animals and livestock in sheep units was around 1:5 (Table 6).

Based on estimates from the 2017 UAV cold season survey results, there were 76,762 Tibetan sheep, 53,365 domestic yaks, and 879 horses in the YRSNP. The YRSNP contains the Huanghe Township, Zhalinghu Township, and Mazhali Town in Madoi county. As the 2017 livestock data in these three townships/towns were missing, the 2015 livestock statistical data were used for comparison and analysis with the UAV survey results. The statistical data provided by the Qinghai Grassland Station showed that there were 73,133 Tibetan sheep, 59,235 domestic yaks, and 1156 horses in the YRSNP in 2015. The results showed that these two sets of data differed only by 4.96%, 9.91%, and 23.96% for Tibetan sheep, domestic yak, and horses, respectively, so the two sets of data were similar.

As the UAV aerial photography resolution for the 2016 warm season is lower (15–18.5 cm), it cannot identify smaller herbivorous animals such as Tibetan sheep and Tibetan gazelles and can only recognize kiangs and domestic yaks. However, populations of kiangs and domestic yaks are easily confused. Therefore, we only estimated the total number of kiangs and domestic yaks in the 2016 warm season survey, which was mainly used for comparison and validation with the 2017 cold season survey. The 2016 warm season UAV survey identified a total of 71,589 individuals for domestic yak and kiangs, whereas the 2017 cold season UAV survey found a total of 66,263 individuals for domestic yak and kiangs, and the two values differed by 7.44%.

### 3.2. Distribution of Forage Yield

#### 3.2.1. Verification of Downscaling Forage Yield Data

In order to better reflect the yield distribution status at the county level, the data fusion method was used to downscale the 500-m spatial resolution forage yield data that were generated by empirical models to a resolution of 30 m. The results clearly reflected the detailed characteristics of forage yield in the YRSNP. Before downscaling, the data could only reflect the rough distribution characteristics of forage yield in YRSNP. After downscaling, the transition between spatial data became smoother and we were therefore able to perform quantitative statistical analysis of the forage yield of different functional zones at a lower scale. In addition, there were no significant differences in the mean forage yield before and after downscaling. From this, we can see that the downscaled data could precisely reflect the spatial differences and variation patterns of forage yield as well as significantly increase the data precision in regions at a low scale.

We compared the downscaled 2016 forage yield data with 48 field forage yield measurement data in the 2016 field sampling survey. The results showed that a significant linear relationship existed between the two datasets (R^2^ = 0.75, *P* < 0.01) (Figure 5). Although the simulated YRSNP forage yield data in this paper and the field sampling data exhibited a good linear relationship, there were still some systematic errors, as indicated by a certain distance between the fitted line and the 1:1 line. This may have been caused by differences in the sampling scale and simulation scale [44].

#### 3.2.2. Forage Yield Distribution in Different Functional Zones

In this paper, empirical models and downscaling were used to estimate the 2016 YRSNP average forage yield. Our findings show that the forage yield per unit area in the entire park was 354.98 kg/hm^2^ and annual total forage yield was 606,800 t. With regard to the type of grassland, the major grassland types in YRSNP include alpine steppes, alpine meadows, and swamps. Among these grasslands, swamps had the highest forage yield per unit area at 509.58 kg/hm^2^, while the forage yields per unit area for alpine meadows and alpine steppes were 401.96 and 229.74 kg/hm^2^, respectively. From this, we can see that forage yield was primarily determined by hydrothermal conditions.

Spatially, the differences between the different regions were extremely large and demonstrated a successive decline from south to north (Figure 6). The forage yield per unit area for the core reserve areas was 340.82 kg/hm^2^, and the total annual forage yield was 248,600 t, accounting for 40.97% of the total annual forage yield. The forage yield per unit area for the traditional utilization areas was 417.91 kg/hm^2^, and the total annual forage yield was 307,700 t, accounting for 50.71% of the total annual forage yield. The forage yield per unit area for ecological restoration areas was 207.40 kg/hm^2^, and the total annual forage yield was 50,500 t, accounting for 8.32% of the total annual forage yield.

### 3.3. Effect of Functional Zones on the Forage–Livestock Balance

#### 3.3.1. Preferences of Large Wild Herbivores for Functional Zones

The core reserve areas (CRAs), traditional utilization areas (TUAs), and ecological restoration areas (ERAs) in the flight samples under UAV survey were 76.05, 97.66, and 34.47 km^2^, respectively. Kiangs and Tibetan gazelles differed in their preference for functional zones, as seen in Table 7.

The frequency of kiang appearances in the core reserve areas was highest among all functional zones (52.78%), but the selection index was relatively low (E*_i_* =0.06), which appeared as random selection in the core reserve areas. This was followed by ecological restoration areas (35.32%) with a high selection index (E*_i_* = 0.25), showing that kiangs have some preference for ecological restoration areas. Conversely, traditional utilization areas accounted for the lowest proportion, with only 11.90% of kiangs being located in this functional zone. In addition, the selection index was the lowest (E*_i_* = −0.67), presenting as negative selection.

The frequency of Tibetan gazelle appearances in the core reserve areas was highest among all functional zones (42.11%), but the selection index was relatively low (E*_i_* = 0.01), which appeared as random selection in the core reserve areas. This was followed by traditional utilization areas (31.58%) with the lowest selection index (E*_i_* = −0.26), indicating that Tibetan gazelles have some negative preference for traditional utilization areas. The ecological restoration areas accounted for the lowest proportion, with only 26.32% of Tibetan gazelles being located in this functional zone; in addition, the selection index was the lowest (E*_i_* = 0.17), presenting as positive selection.

These findings indicate that the two large wild herbivorous animals positively select ecological restoration areas. The vegetation type in this region type consists of degraded and desertified grassland that urgently requires restoration and has low forage yield values. The choice of this habitat is primarily determined by the characteristics of the two animals, who are adept at running, and thus the flat desertified grassland satisfies this requirement (these animals do not have stringent water supply requirements). These findings are consistent with the study of Dong et al. [45] and Wu et al. [46], who assessed the habitat preferences of kiangs in the Aerjin Mountain Nature Reserve and found that kiangs prefer alpine deserts or desert steppes with low vegetation cover.

#### 3.3.2. Forage–Livestock Balance based on Large Herbivores

Based on the forage yield data following downscaling and relevant parameter calculations, we discovered that the theoretical livestock carrying capacity per unit area of grassland in YRSNP in 2016 was 0.211 sheep units/hm^2^. The theoretical livestock carrying capacities per unit area of grassland in the core reserve areas, traditional utilization areas, and ecological restoration areas were 0.202, 0.248, and 0.123 sheep units/hm^2^, respectively, showing an increasing trend from the north to south (Figure 7).

From the entire region, we can see that the areas in which the forage–livestock balance were maintained totaled 103,00 km^2^, accounting for 60.30% of the grassland area in the entire region. These regions were mainly located at core reserve areas and ecological restoration areas. The areas of slight and moderate overloading were 700 and 900 km^2^, respectively, accounting for 3.93% and 5.09% of the grassland area in the entire region, respectively. These regions were mainly located at the south and west of YRSNP. The areas of heavy and extreme overloading were 500 and 4700 km^2^, respectively, accounting for 2.96% and 27.73% of the grassland area in the entire region, respectively. These regions were mainly located at the traditional utilization areas (Figure 8).

The Three-River Source National Nature Reserve Masterplan clearly specifies that grazing activities are prohibited in core reserve areas and ecological restoration areas, in order to maintain the health and stability of alpine wetland ecosystems and conserve the authenticity and integrity of natural landscapes. Highly stringent grassland protection measures have thus been implemented, and for this reason we only considered the pressure caused by kiangs and Tibetan gazelles on grasslands in this paper. From the calculations using Equations (6) and (7) based on the actual livestock carrying capacity per unit area grassland for wild animals (0.034 sheep units/hm^2^), the grazing pressure indices of the core reserve areas and ecological restoration areas were 0.168 and 0.276, respectively, showing that these two regions still possess high livestock carrying capacities.

For traditional utilization areas, the pressures caused by livestock and wild animals on grasslands must be considered. The actual livestock carrying capacity per unit area was 0.435 sheep units/hm^2^ and the grazing pressure index was 1.754, indicating that the grassland was heavily overloaded (Table 8). In order to maintain the forage–livestock balance, there is a need to coordinate the population and carrying capacity of the natural environment. Therefore, further livestock reduction is required. Based on the premise of ensuring a stable number of wild herbivorous animals, there is a need for a 45% reduction in livestock to ensure the forage–livestock balance in traditional utilization areas, which is around 132,400 sheep units.

The partitioning of functional zones in national parks has a significant effect on the forage–livestock balance in YRSNP. Before the delineation of functional zones that were based on the actual livestock carrying capacity per unit area of grassland for livestock and large wild herbivores in YRSNP (0.207 sheep units/hm^2^), the grazing pressure index was 0.981. Following the implementation of the plan (and without reducing the number of livestock), the actual livestock carrying capacity was 0.415 sheep units/hm^2^ and the grazing pressure index was 1.967. Grazing pressure was increased 1-fold, and the forage–livestock conflict became more pronounced.

## 4. Discussion

Conventional forage–livestock balance research has mainly focused on livestock feed intake, pasture yield, grassland availability, proportion of edible pastures, and livestock carrying capacity [47,48,49]. It has only used statistical data of livestock to estimate the actual livestock carrying capacity. However, wild herbivorous animals have a large body size, with an individual feed intake that is not less than cows, sheep, or other livestock; for instance, the daily feed intake of one kiang is equivalent to four sheep units. Therefore, if wild herbivorous animals are not included in the calculations of grazing pressure, the actual grazing pressure of grasslands can be easily underestimated. Yang et al. [50] found that grazing pressure was underestimated by 22% when the number of wild herbivorous animals was not considered in studies in Madoi county in the Three-Rivers Source region.

However, large wild herbivores usually avoid humans and have low densities, so uncertain activity ranges and are easily concealed and it can be difficult to obtain accurate and real-time information on population numbers. UAV platforms not only compensate for this deficiency, but help overcome the limitations of conventional remote sensing technology at temporal and spatial resolutions. These platforms can directly monitor animal data and distributions, demonstrating unique advantages in animal diversity studies, particularly surveys on the population numbers of large animals and animal conservation. Despite these advantages, UAV remote sensing still has some limitations in animal diversity research. For instance, the load carried by UAVs is low, and it is difficult to integrate multiple sensors in the same platform for observation, which is currently still primarily based on optical cameras. Due to battery capacity limitations, UAVs can only stay in the air for a short period of time, and thus are unable to perform long-term monitoring of wild animals. In addition, the area monitored is smaller than other remote sensing platforms. In contrast to areas with unobstructed views, such as grasslands, it is difficult to directly obtain species information in forests [51].

Over the last 60 years, the number of livestock in the Tibetan Plateau has increased by around five-fold, peaking in the 1970s and showing a slight decrease in the 1990s [52]. However, the base number of livestock is still large (Figure 9). Since the implementation of the Three-River Source Region Ecological Project in 2005, the number of livestock in Madoi county has decreased from 608,000 sheep units to 389,000 sheep units, corresponding to a significant livestock reduction rate of 36%. However, the Three-River Source National Nature Reserve Masterplan clearly stated that the number of wild animal populations should be increased by 20% in 2020 while maintaining the forage–livestock balance. This means that competition between large populations of wild herbivores and domestic animals for resources in YRSNP will increase, and thus grazing pressure will also increase. Therefore, livestock grazing management still constitutes a major issue. In this survey, the UAV flew 191.11 km^2^ in core reserve areas such as the Gyaring Lake, Ngoring Lake, and Xingxing Sea, where it found 4389 sheep units of livestock, with a density of 22.97 sheep units/km^2^. This differed only slightly from the livestock density in non-core reserve areas (25.78 sheep units/km^2^). This shows that although core reserve areas do not allow grazing and other human activities, they still contain large numbers of livestock, which directly reduces the available space for wild animals.

The grassland animal husbandry industry, which mainly involves grazing, is a key factor influencing the functioning of alpine steppes. Overgrazing and seasonal imbalances caused by traditional animal husbandry severely threaten the maintenance of ecosystem services [53,54]. There is a need therefore for tailored conservation management measures for the three different functional zones. For example, compensation should be paid to herders for losses associated with livestock reduction in traditional utilization areas as a result of wildlife conservation. In addition, the duration of livestock grazing should be reduced, and the seasonal suspension of grazing and rotational grazing should be implemented. The appropriate development of new intelligent ecological animal husbandry models with a goal of “reducing pressure and increasing efficiency” could be used to increase the economic value of livestock. An emphasis on the maintenance of natural ecological processes should be carried out in core reserve areas. Strict prohibition measures should be adopted to limit and reduce various types of human activities. Monitoring of wildlife and their habitats should be strengthened, and changes in animal populations and quantity should be closely tracked. Ecological restoration areas should focus on natural restoration and should develop necessary artificial interventions to accelerate the restoration of degraded grasslands and strengthen ecological monitoring and periodic evaluation.

Various limitations were noted in the present study. During the 2016 warm season, the resolution of the UAV aerial photography was only 15–18.5 cm, and only larger animals could be identified. In addition, distinguishing between herds of kiangs and domestic yaks proved challenging. In the 2017 cold season, the high degree of overlap between the UAV aerial photography images influenced the flight efficiency, which increased the image processing workload. When UAVs are used for surveys of large herbivores in the future, the image resolution should be increased as much as possible (optimal: 4–5 cm), while flight efficiency and fore-and-aft overlap and side overlap should be reduced as much as possible. With regards to image information extraction, HCI interpretation was only used to obtain the number and location of animals, and animal dimensions and other information were not accurately extracted. In the later stages, the characteristic morphology and biological characteristics of animals in the UAV images were considered, and computerized automatic recognition algorithms were used to achieve the automatic interpretation of animals, and to extract animal size and other characteristics to deduce the internal population structure at the species-level. In addition, wild animals migrate, and thus the monitoring of wild animals is subject to spatiotemporal effects. Conventional large-scale monitoring can reveal the variation patterns of population numbers and the associated driving mechanisms. Therefore, the number of wild animal populations should be continuously monitored. Furthermore, the role predators have played in the dynamics of these wild and domestic populations remains to be further studied [55,56].

## 5. Conclusions

In this paper, UAV flight sample strip survey results and pasture distribution data of cold and warm seasons in YRSNP were used to estimate the numbers of large herbivorous animal populations and analyze the effects of functional zone delineation in national parks on the forage–livestock balance. 

Aerial photography surveys of large herbivorous animals in YRSNP were conducted under the guidance of the Technical Guidelines for the 2nd National Survey of Terrestrial Wildlife Resources and the National Terrestrial Wildlife Resource Survey and Monitoring Technical Guidelines issued by the State Forestry and Grassland Administration. After image stitching and HCI interpretation, large herbivores were identified in the UAV aerial photography sample strip regions in the cold season. The density of kiangs and Tibetan gazelles obtained by the UAV surveys was generally consistent with the line transect survey method. The density of domestic yaks, Tibetan sheep, and horses was accurately investigated.

The type and quantity of large herbivores obtained by the UAV sample strip surveys were combined with data on the characteristics of the wild animals and cold and warm season pasture distribution data. This was done for spatial statistical analysis and to achieve a scientific estimate of the relative number of individuals based on the regional inference of the absolute number of individuals in the sample strips. This was used for comparison and validation with the statistical data and warm season survey results. The numbers of kiangs, Tibetan gazelles, domestic yaks, Tibetan sheep, and horses were estimated, and the total number of sheep units for large wild herbivores and livestock was calculated, with the ratio of large wild herbivores to livestock by sheep units being 1:5.

Large wild herbivores have different preferences for functional zones, positively selecting ecological restoration areas. These areas consist mainly of degraded grassland with low forage yields. Therefore, there is a need to strengthen the monitoring of wild animals and their habitats in these areas, conduct periodic evaluations, implement strict forage–livestock balance regulations, carry out seasonal suspensions of grazing and rotational grazing, and implement remediation for severely degraded grasslands.

The partitioning of functional zones in national parks has significant effects on the forage–livestock balance in YRSNP. After the implementation of the plan, and without reducing the number of livestock, there still existed an overloading of grassland in YRSNP. Grazing pressure increased one-fold, and forage–livestock conflict became more pronounced. Long-term mechanisms for the forage-livestock balance could promote grassland ecological improvement. Under the premise of ensuring stable numbers of wild herbivorous animals, a livestock reduction of 45% is required to ensure forage–livestock balance in the national park. In contrast to a reduction, other methods such shifting use to other grazing areas could be taken into consideration.

## Figures and Tables

**Figure 1 ijerph-16-04402-f001:**
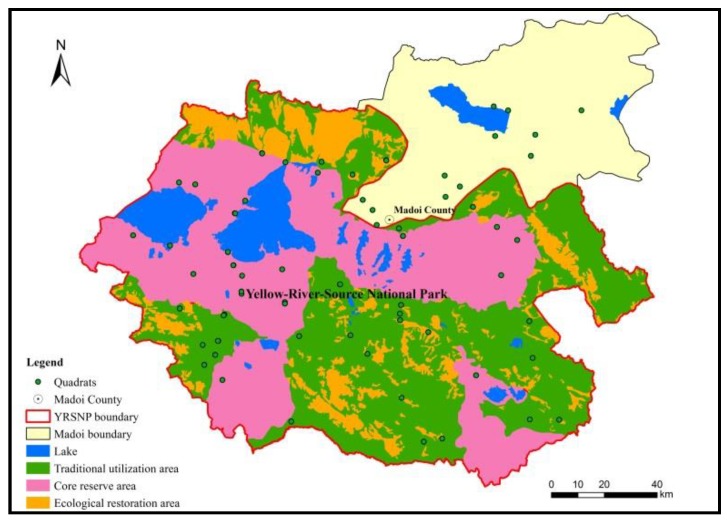
Functional zone of the Yellow-River-Source National Park (YRSNP).

**Figure 2 ijerph-16-04402-f002:**
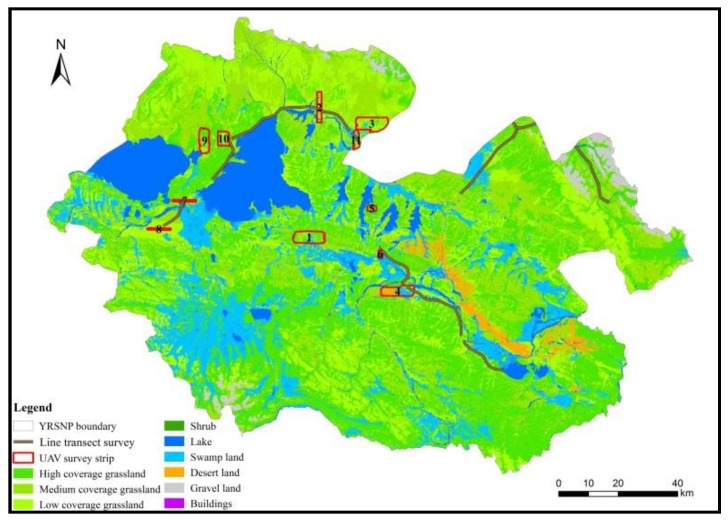
Unmanned aerial vehicle survey and line transect survey during the winter season in 2017.

**Figure 3 ijerph-16-04402-f003:**
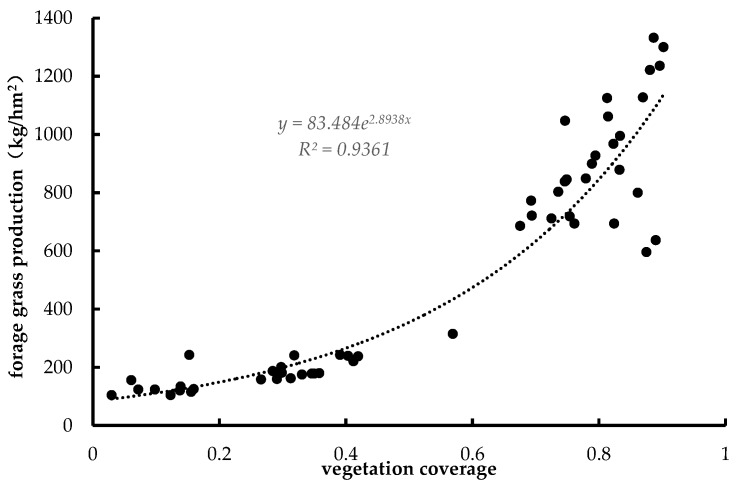
Downscaled fitting curve of forage grass production.

**Figure 4 ijerph-16-04402-f004:**
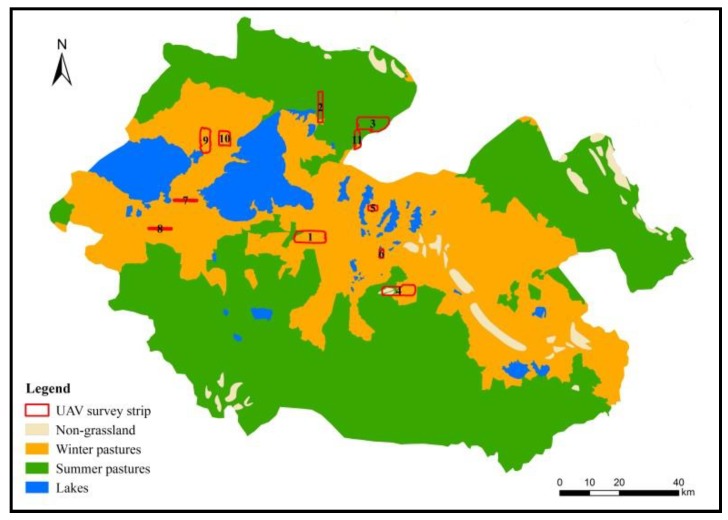
Season grassland distribution map of Madoi County.

**Figure 5 ijerph-16-04402-f005:**
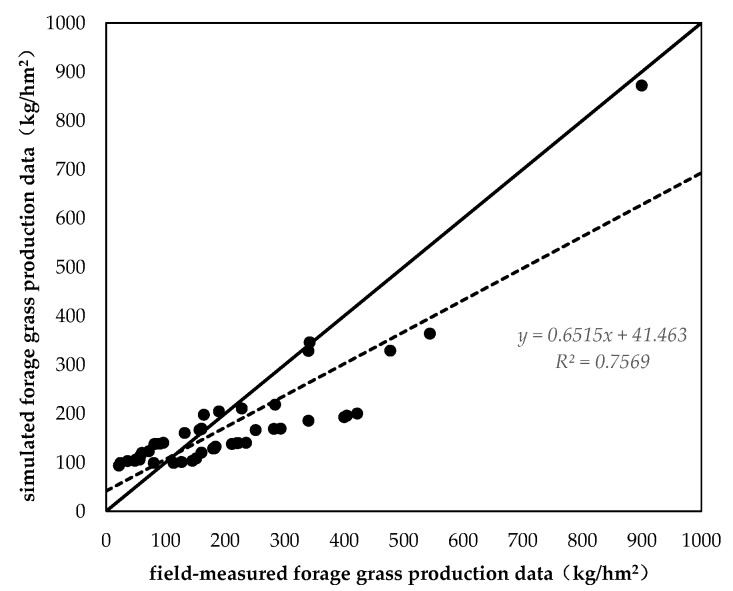
Accuracy verification of downscaling forage yield data.

**Figure 6 ijerph-16-04402-f006:**
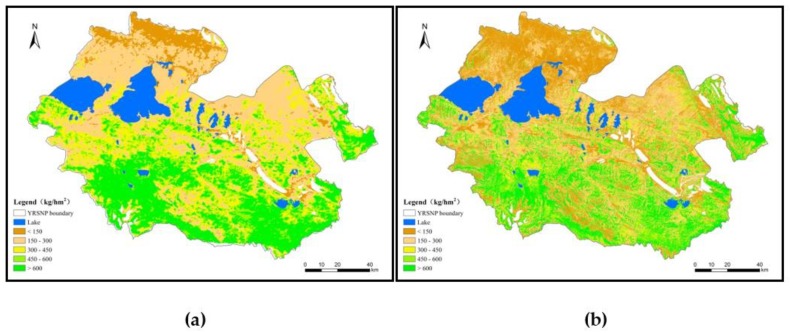
Spatial distribution of forage yield in the YRSNP in 2016: (**a**) before downscaling; (**b**) after downscaling.

**Figure 7 ijerph-16-04402-f007:**
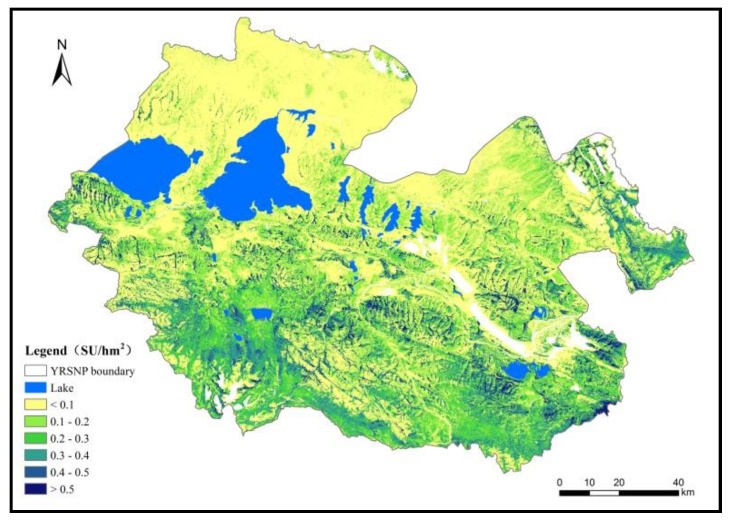
Spatial distribution of the theoretical number of livestock in YRSNP.

**Figure 8 ijerph-16-04402-f008:**
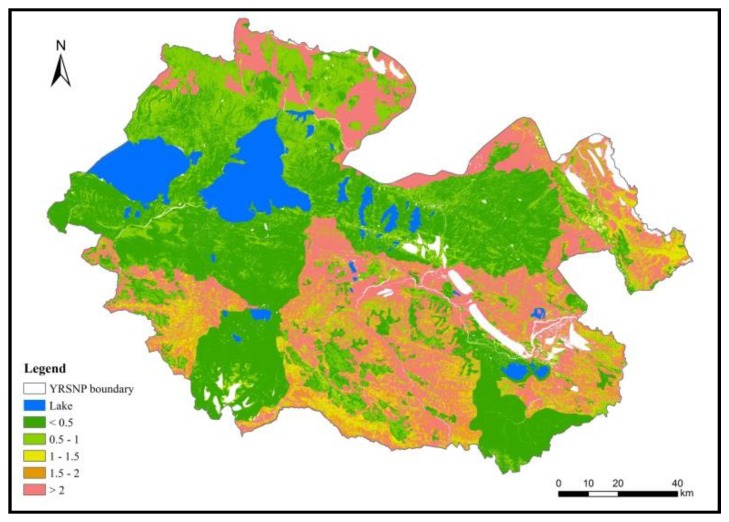
Spatial distribution of grazing pressure in YRSNP (no unit).

**Figure 9 ijerph-16-04402-f009:**
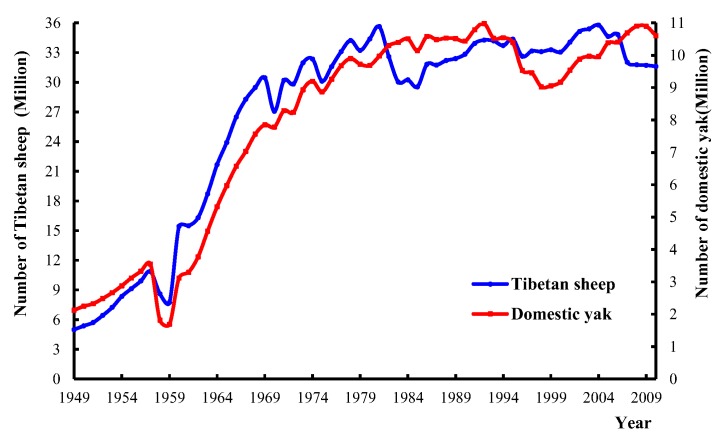
Variation curve of livestock numbers in the last 60 years in the Tibetan Plateau.

**Table 1 ijerph-16-04402-t001:** Empirical models of different types of grassland.

Type of Grassland	Type of Empirical Model	Equation	R^2^	RMSE
Alpine meadow	Linear	*Y = 6348.8NDVI-3189.1*	0.425	516.17
	logarithmic	*Y = 4296ln(NDVI) + 2823.3*	0.434	551.45
	power	*Y = 3798.9NDVI^3.537^*	0.422	541.54
	exponential	*Y = 27.574e^5.1936NDVI^*	0.436	515.86
	quadratic polynomial	*Y = 18891NDVI^2^-20170NDVI + 5985.6*	0.430	540.29
Alpine steppe	Linear	*Y = 1078.4NDVI-64.511*	0.489	197.89
	logarithmic	*Y = 406.32ln(NDVI) + 776.32*	0.460	195.44
	power	*Y = 944.25NDVI^1.145^*	0.527	205.36
	exponential	*Y = 89.993e^2.998NDVI^*	0.545	185.70
	quadratic polynomial	*Y = 451.27NDVI^2^ + 686.69NDVI + 8.72*	0.490	200.72

**Table 2 ijerph-16-04402-t002:** Optimal empirical models of different types of grassland.

Index Level	0–0.96	0.97–1.03	1.04–1.25	1.26–1.65	1.66–1.99	>2
Index description	With grazing potential	Basic balance	Slightly overloaded	Moderately overloaded	Heavily overloaded	Extremely overloaded

**Table 3 ijerph-16-04402-t003:** Standard sheep unit conversion table for each herbivorous animal.

Type of Animal	Kiang	Tibetan Gazelle	Domestic Yak	Tibetan Sheep	Horse
Sheep unit	4	0.5	4	1	6

**Table 4 ijerph-16-04402-t004:** Density statistics of large herbivorous animal sample strips in YRSNP in spring 2017.

ID of Sample Strip	Animal Species	Total Number	Density (individual/km^2^)	Density (SU/km^2^)
1	Kiang	107	2.66	10.63
	Tibetan gazelle	23	0.57	0.29
	Domestic yak	639	15.86	63.46
	Tibetan sheep	4	0.10	0.10
	Horse	8	0.20	1.19
	Total	781	19.39	75.66
2	Kiang	6	0.39	1.54
	Tibetan gazelle	23	1.48	0.74
	Total	29	1.86	2.28
3	Kiang	4	0.09	0.36
	Tibetan gazelle	10	0.23	0.11
	Total	14	0.32	0.48
4	Kiang	2	0.06	0.23
	Total	2	0.06	0.23
No animal was found in No. 5 sample strip				
6	Horse	2	0.50	2.99
	Total	2	0.50	2.99
7	Domestic yak	1	0.22	0.89
	Total	1	0.22	0.89
No animal was found in No. 8 sample strip				
9	Kiang	48	1.84	7.34
	Domestic yak	6	0.23	0.92
	Tibetan sheep	891	34.06	34.06
	Horse	4	0.15	0.92
	Total	949	36.28	43.24
10	Kiang	85	4.75	18.99
	Tibetan gazelle	1	0.06	0.03
	Domestic yak	217	12.12	48.49
	Total	303	16.93	67.51
11	Domestic yak	167	15.11	60.46
	Total	167	15.11	60.46

**Table 5 ijerph-16-04402-t005:** Wild animal line transects results.

ID of Line	Length of Line (km)	Number of Kiang	Number of Tibetan Gazelle	Density of Kiang (individual/km^2^)	Density of Tibetan Gazelle (individual/km^2^)
1	1.62	112	0	69.14	0.00
2	14.32	0	0	0.00	0.00
3	79.72	25	25	0.31	0.31
4	21.16	11	11	0.52	0.52
5	53.43	5	58	0.09	1.09
6	1.56	0	0	0.00	0.00
7	1.35	5	0	3.70	0.00
8	37.88	23	12	0.61	0.32
9	23.4	52	24	2.22	1.03
10	33.71	1	2	0.03	0.06
11	35.94	2	0	0.06	0.00
Total	304.09	236	132	0.78	0.43

**Table 6 ijerph-16-04402-t006:** Population number of large herbivores of winter season.

Animal Species	Grassland Type	Density (individuals/km^2^)	Area (km^2^)	Total Number	Sheep Unit (×10^4^)
kiang	Winter pasture	1.71	6841.56	11699	4.68
	Summer pasture	0.12	9992.72	1199	0.48
	Total	0.77	16834.28	12898	5.16
Tibetan gazelle	Winter pasture	0.53	6841.56	3626	0.18
	Summer pasture	0.85	9992.72	8494	0.42
	Total	0.72	16834.28	12120	0.60
Domestic yak	Winter pasture	5.58	6841.56	38176	15.27
	Summer pasture	1.52	9992.72	15189	6.08
	Total	3.18	16834.28	53365	21.35
Tibetan sheep	Winter pasture	11.22	6841.56	76762	7.68
	Summer pasture	0.00	9992.72	0	0.00
	Total	4.56	16834.28	76762	7.68
Horse	Winter pasture	0.07	6841.56	479	0.29
	Summer pasture	0.04	9992.72	400	0.24
	Total	0.05	16834.28	879	0.53

**Table 7 ijerph-16-04402-t007:** Preferences of large wild herbivores for functional zones.

Animal Species	Functional Zones	Number of Surveyed Quadrats	Number of Selected Quadrats	Selection Coefficient	Selection Index	Selectivity
Kiang	CRA	84541	133	0.38	0.06	RS
	TUA	108484	30	0.07	−0.67	NS
	ERA	38344	89	0.56	0.25	PS
Tibetan gazelle	CRA	84541	24	0.34	0.01	RS
	TUA	108484	18	0.20	−0.26	NS
	ERA	38344	15	0.47	0.17	PS

Note: CRAs = core reserve areas; TUAs = traditional utilization areas; ERAs = ecological restoration areas; RS = random selection; NS = negative selection; PS = positive selection.

**Table 8 ijerph-16-04402-t008:** Preferences of large wild herbivores for functional zones.

Functional Zone	Actual Carrying Capacity (sheep unit/hm^2^)	Theoretical Carrying Capacity (sheep unit/hm^2^)	Grazing Pressure	Forage–Livestock Balance
CRA	0.034	0.202	0.168	With grazing potential
TUA	0.435	0.248	1.754	Heavily overloaded
ESA	0.034	0.123	0.276	With grazing potential
Total	0.415	0.211	1.967	Heavily overloaded

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
