# Peer review of "A Population Census of Large Herbivores Based on UAV and Its Effects on Grazing Pressure in the Yellow-River-Source National Park, China"

_ijerph, 2019, doi:10.3390/ijerph16224402_

Round 1
Reviewer 1 Report
General assessment:
The paper provides interesting information in the field of remote sensing as a tool of large herbivores management. The subject falls within the general scope of the International Journal of Environmental Research and Public Health. The article demonstrates that drones are useful to determine the herbivore carrying capacity in alpine pastures. The interpretations are well justified and consistent with the objectives. The discussion is very focused on the study area (the national park). It is necessary to give a more global approach to the results. The conclusions must be improved because in large part they are nothing more than results. Listed below are several minor aspects that need to be changed in order to clarify the article. With attention to these points the paper should be acceptable.
Specific points:
- Lines 15-16. Write the scientific names in italics
- Lines 34-35. ‘grazing pressure’ and ‘Yellow-River-Source National Park’ are in the title, so is not necessary they are too in the keywords
- Lines 120-121. Add reference of where this guideline is published
- Lines 128-136. Indicate which camera and lens did you used
- Line 138. HCI is a word with a very broad meaning. More details are required. Did you used machine learning algorithms for recognizing different animal species?
- Lines 142-146. About the procedure, what was the risk of a moving animal appearing in two different images?
- Line 165. That distance, is the average or the maximum?
- Line 168. How was the habitat area established?
- Line 181. A table with areas of different grassland types could be useful
- Lines 186-189. This sentence is confused. Please rewrite.
- Line 216: Write the reference of Vanderloeg and Scavia
- Line 239: If 1.8 kg are dry matter, please write 1.8kgDM/day
- Lines 284-286: Could you explain why there were no animals? Could be they gone after the first survey? If so, how surveys affect their behavior?
- Line 290: studieshave => studies have
- Lines 291-292: Specify the altitude of high elevation
- Line 326: Please, specify these empirical models
- Line 344: Specify if that forage yield is an average
- Lines 363-364: These areas should be drawn in someone of the maps
- Table 7: Write the meaning of RS, NS, PS, CRA, TUA and ERA in the header of the table
- Figura 8: Write the units of grazing pressure at the food of the figure
- Line 413: Specify if 0.034 sheep units/hm2 are per year or per season
- Line 496: UAB => UAV
- Lines 518-521: This paragraph is a result, not a conclusion
- Lines 528-531: idem
- Lines 539-541: idem
- Line 548: Programof => Program of
- Lines 551: forthe => for the
- Line 552: andsuggestions => and suggestions
Author Response
Dear Editors and Reviewers:
Thank you for your letter and for the reviewers’ comments concerning our manuscript entitled “A population census of large herbivores based on UAV and its effects on grazing pressure in the Yellow-River-Source National Park, China” (ID: ijerph-628504). Those comments are all valuable and very helpful for revising and improving our paper, as well as the important guiding significance to our study. We have studied comments carefully and have made correction which we hope meet with approval. Revised portion are marked in red in the paper. The main corrections in the paper and the responds to the reviewer’s comments are as following:
Responds to the reviewer’s comments:
Lines 15-16: Write the scientific names in italics.Response: We have re-written the scientific names in italics according to the Reviewer’s suggestion. Please see lines 15-16 on page 1.
Lines 34-35: ‘grazing pressure’ and ‘Yellow-River-Source National Park’ are in the title, so is not necessary they are too in the keywords.Response: We have made correction according to the Reviewer’s comments. Please see lines 34-35 on page 1.
Lines 120-121: Add reference of where this guideline is published.Response: We have added reference of the published guideline according to the Reviewer’s comments. Please see lines 628-630 on page 19.
Lines 128-136: Indicate which camera and lens did you used.Response: We have added model of camera and lens according to the Reviewer’s comments. Please see line 131 on page 4.
Line 138: HCI is a word with a very broad meaning. More details are required. Did you used machine learning algorithms for recognizing different animal species?Response: Human–computer interaction (HCI) method means that the professional interpreters in the aspect of remote sensing and herbivores identify the message of remote sensing using geographic information system software. HCI was only used to obtain the number and location of animals, and animal dimensions and other information were not accurately extracted. In next work, the characteristic morphology and biological characteristics of animals in the UAV images were considered, and machine learning algorithms or computerized automatic recognition algorithms were used to achieve the automatic interpretation of animals and to extract animal size and other characteristics to deduce the internal population structure.
Lines 142-146: About the procedure, what was the risk of a moving animal appearing in two different images?Response: The speed of UAVs is much faster than the movement speed of animal, so the animal will not appear in two different images, and the low degree of overlap ensures that the same animal will not be counted repeatedly.
Line 165: That distance, is the average or the maximum?Response: W is the unilateral average observation distance. Please see line 168 on page 5.
Line 168: How was the habitat area established?Response: where S is the total habitat area of the target animal species (except for lakes and buildings). We have made correction according to the Reviewer’s comments. Please see line 171 on page 5.
Line 181: A table with areas of different grassland types could be usefulResponse: “grassland type” should be “grassland utilization type”. There are two types: cold season pastures and warm season pastures.
Lines 186-189: This sentence is confused. Please rewrite.Response: We have re-written this sentence according to the Reviewer’s suggestion. Please see lines 186-193 on page 5.
Line 216: Write the reference of Vanderloeg and Scavia.Response: We have made correction according to the Reviewer’s comments. Please see line 220 on page 6.
Line 239: If 1.8 kg are dry matter, please write 1.8kgDM/day.Response: We have made correction according to the Reviewer’s comments. Please see line 243 on page 7.
Lines 284-286: Could you explain why there were no animals? Could be they gone after the first survey? If so, how surveys affect their behavior?Response: The survey area is selected randomly every day to ensure the reliability of the results. No animals were found by ground synchronous surveys or UAV surveys in 15 and 16 April. The UAV survey results were the same as the ground survey results. This shows UAV survey method has a good accuracy and feasibility.
Line 290: studieshave => studies haveResponse: We have made correction according to the Reviewer’s comments. Please see line 294 on page 9.
Lines 291-292: Specify the altitude of high elevationResponse: We have made correction according to the Reviewer’s comments. Please see line 295 on page 9.
Line 326: Please, specify these empirical modelsResponse: In practice, the number of observations is always finite, and the true value can only be the most reliable value instead. The RMS error (RMSE) is very sensitive to large or very small error in a set of measurements. Therefore, the RMSE can well reflect the precision of the measurement. RMSE reflects the degree of deviation of the measured data from the true value. The smaller RMSE means higher measurement accuracy. Therefore, RMSE can be used as a criterion for evaluating the accuracy of this measurement process.
Figure1.Downscaling fitting curve of forage grass production.
Supplementary explanation: (Figure1.Downscaling fitting curve of forage grass production.)
|
Type of curve fitting |
equation |
R2 |
RMSE |
|
|
Linear |
Y=1252.2VC-122.58 |
0.843 |
155.97 |
|
|
logarithmic |
Y=384.36Ln(VC)+882.76 |
0.630 |
239.34 |
|
|
power |
Y=884.69VC0.9318 |
0.770 |
202.23 |
|
|
exponential |
Y=83.484e2.8938VC |
0.936 |
137.20 |
|
|
quadratic polynomial |
Y=1555.2VC2-302.72VC+133.95 |
0.889 |
138.88 |
|
Response: Empirical models and downscaling were used to estimate the YRSNP average forage yield. Please see line 348 on page 11.
Lines 363-364: These areas should be drawn in someone of the mapsResponse: These areas have been drawn in figure 1. Please see figure1on page 3.
Table 7: Write the meaning of RS, NS, PS, CRA, TUA and ERA in the header of the tableResponse: We have made correction according to the Reviewer’s comments. Please see lines 370-371 on page 12.
Figure 8: Write the units of grazing pressure at the food of the figureResponse: We have made correction according to the Reviewer’s comments. Please see line 411 on page 14.
Line 413: Specify if 0.034 sheep units/hm2 are per year or per seasonResponse: The actual livestock carrying capacity per year was 0.435 sheep units/hm2. Please see line 422 on page 14.
Line 496: UAB => UAVResponse: We have made correction according to the Reviewer’s comments. Please see line 501 on page 16.
Lines 518-521: This paragraph is a result, not a conclusion; Lines 528-531: idem; Lines 539-541: idemResponse: We have re-written these sentences according to the Reviewer’s suggestion. Please see lines 525-557 on page 17.
Line 548: Programof => Program ofResponse: We have made correction according to the Reviewer’s comments. Please see line 553 on page 17.
Lines 551: forthe => for theResponse: We have made correction according to the Reviewer’s comments. Please see line 556 on page 17.
Line 552: andsuggestions => and suggestionsResponse: We have made correction according to the Reviewer’s comments. Please see line 557 on page 17.
We tried our best to improve the manuscript and made some changes in the manuscript. These changes will not influence the content and framework of the paper. And here we did not list the changes but marked in red in revised paper.
We appreciate for Editors/Reviewers’ warm work earnestly, and hope that the correction will meet with approval.
Once again, thank you very much for your comments and suggestions.

Reviewer 2 Report
I enjoyed reading this article. The methods and results are well-written and substantial contributions to the field. I would have enjoyed a little more discussion of the ecology of the herbivores under study. I also would have liked to learn about the predator populations in this system and how they could impact the forage-livestock balance and interpretation of your results if considered in your approach.
I have attached a pdf with comments. I have also highlighted areas in need of grammatical attention.
Some potential papers to help introduce the ecology of these herbivores and predator prey dynamics:
Resource selection in a high-altitude rangeland equid, the kiang (Equus kiang): influence of forage abundance and quality at multiple spatial scales
Antoine St-Louis, Steeve D. Côté
» Abstract
Canadian Journal of Zoology, 2014, 92:239-249, https://doi.org/10.1139/cjz-2013-0191
Snow Leopard and Himalayan Wolf: Food Habits and Prey Selection in the Central Himalayas, Nepal
Chetri M, Odden M, Wegge P (2017) Snow Leopard and Himalayan Wolf: Food Habits and Prey Selection in the Central Himalayas, Nepal. PLOS ONE 12(2): e0170549. https://doi.org/10.1371/journal.pone.0170549

Author Response
Dear Editors and Reviewers:
Thank you for your letter and for the reviewers’ comments concerning our manuscript entitled “A population census of large herbivores based on UAV and its effects on grazing pressure in the Yellow-River-Source National Park, China” (ID: ijerph-628504). Those comments are all valuable and very helpful for revising and improving our paper, as well as the important guiding significance to our study. We have studied comments carefully and have made correction which we hope meet with approval. Revised portion are marked in red in the paper. The main corrections in the paper and the responds to the reviewer’s comments are as following:
Responds to the reviewer’s comments:
Line 21: statistical data?Response: “statistical data” should be “official statistics”. Please see line 21 on page 1.
Line 27: explain degraded state, use a period not a semi-colon.Response: We have made correction according to the Reviewer’s comments. Please see lines 27 on page 1.
Line 30: Use a period not a semi-colon.Response: We have made correction according to the Reviewer’s comments. Please see lines 30 on page 1.
Line 33: The abstract could use some better transitions and fuller or complete messages for each point. It does not flow from sentence to sentence. Does the concluding sentence refer to global forage-livestock conflict or those only in Chana, for example?Response: We are very sorry for our unclear expression. Forage–livestock conflict has become more severe in YRSNP. Please see line 33 on page 1.
Line 41: activation of what?Response: We have deleted this word “region and important activation” to make sentence clear. Please see lines 40-41 on page 1.
Line 46: This place would be an ideal location for an example of this conflict.Response: We have added this sentence in this paragraph. Please see lines 46-47 on page 2.
Line 59: Could explain the differences in the amount of time each method consumes to provide an estimate of population size for an equal spatial area?Response: Ground surveys have high costs, are time-consuming, and cannot be frequently conducted in a large area. In terms of herbivore population size estimation, time cost of UAV surveys is far lower than that of ground surveys. Please see lines 62-63 on page 2.
Line 78: This whole paragraph reads like a list of studies and their brief summaries. This Fan et al. study could at least use a better explanation of approach and findings, which would entail 2-3 sentences.Response: Fan et al. used the Global Production Efficiency Model (GLO-PEM) model to calculate forage yield and grazing pressure, results indicated that climate change is the main factor affecting forage yield in the Three-River Source region, and different grazing management models similarly affect forage yield. Please see lines 78-79 on page 2.
Line 152: the species of wild animals?Response: We have made correction according to the Reviewer’s comments. Please see line 158 and 167.
Line 170: Could write-out HCI for its first entry in the ms?Response: Human–Computer Interaction (HCI). Please see line 176 on page 5.
Line 176: This statement needs a citation to bolster it.Response: Joseph, L.; Bard-Jorgen B. Density of Tibetan antelope, Tibetan wild ass and Tibetan gazelle in relation to human presence across the Chang Tang Nature Reserve of Tibet, China. Acta Zool. Sin. 2005, 51:586-597. We have made correction according to the Reviewer’s comments. Please see line 182 on page 5.
Line 193: Citation for Qinghai grassland terminal?Response: We have deleted this sentence according to the Reviewer’s comments. Please see line 200 on page 7.
Table 6: Why numbers and not the actual species name?Response: We have made correction according to the Reviewer’s comments. Please see line 311 on table 6.
Table 7: Readers should be reminded about definitions of RS, PS, NS near here.Response: CRA = core reserve areas; TUA = traditional utilization areas; ERA = ecological restoration areas; RS = random selection; NS = negative selection; PS= positive selection. Please see lines 373-374 on table 7.
Line 446: This sentence is long and difficult to understand. Break apart into two sentences.Response: However, large wild herbivores usually avoid humans, they have low density, uncertain activity ranges and are easily concealed, so it is difficult to obtain accurate and real-time information on population numbers. Please see lines 454-455 on page 15.
Line 461: What role have predators played in the dynamics of these wild and domestic populations?Response: What role have predators played in the dynamics of these wild and domestic populations remains to be further studied. Please see lines 518-519 on page 16.
We tried our best to improve the manuscript and made some changes in the manuscript. These changes will not influence the content and framework of the paper. And here we did not list the changes but marked in red in revised paper.
We appreciate for Editors/Reviewers’ warm work earnestly, and hope that the correction will meet with approval.
Once again, thank you very much for your comments and suggestions.
